# Structure–Properties Relationship in Waterborne Poly(Urethane-Urea)s Synthesized with Dimethylolpropionic Acid (DMPA) Internal Emulsifier Added before, during and after Prepolymer Formation

**DOI:** 10.3390/polym12112478

**Published:** 2020-10-26

**Authors:** Mónica Fuensanta, Abbas Khoshnood, José Miguel Martín-Martínez

**Affiliations:** Adhesion and Adhesives Laboratory, University of Alicante, 03080 Alicante, Spain; monica.fuensanta@ua.es (M.F.); abbas.khoshnood@ua.es (A.K.)

**Keywords:** waterborne poly(urethane-urea), dimethylolpropionic acid (DMPA), methyl ethyl ketone (MEK) prepolymer method, structure–properties relationship, micro-phase separation, hard and soft segments, thermal properties, adhesion, viscoelastic properties

## Abstract

Dimethylolpropionic acid (DMPA) internal emulsifier has been added before, during and after prepolymer formation in the synthesis of waterborne poly(urethane-urea)s (PUDs) and their structure–properties relationships have been assessed. PUDs were characterized by pH, viscosity and particle size measurements, and the structure of the poly(urethane-urea) (PU) films was assessed by infra-red spectroscopy, differential scanning calorimetry, X-ray diffraction, thermal gravimetric analysis, plate–plate rheology and dynamic mechanical thermal analysis. The adhesion properties of the PUDs were measured by cross-hatch adhesion and T-peel test. The lowest pH value and the highest mean particle size were found in the PUD made by adding DMPA *after* prepolymer formation, all PUDs showed relatively ample mono-modal particle size distributions. The highest viscosity and noticeable shear thinning were obtained in the PUD made by adding DMPA *during* prepolymer formation. Depending on the stage of addition of DMPA, the length of the prepolymer varied and the PU films showed different degree of micro-phase separation. Because the shortest prepolymer was formed in the PU made with DMPA added *before* prepolymer, this PU film showed the lowest storage moduli and early melting indicating higher degree of micro-phase separation. The highest storage modulus, later melting, higher temperature and lower modulus at the cross between the storage and loss moduli corresponded to the PU made by adding DMPA *after* prepolymer formation, because the longer prepolymer produced during synthesis. The lowest thermal stability corresponded to the PU made by adding DMPA *during* prepolymer formation and the structures of all PU films were dominated by the soft domains, the main structural differences derived from the hard domains. Whereas DMPA-isophorone diisocyanate (IPDI) urethane and urea hard domains were created in the PU film made by adding DMPA *during* prepolymer formation, the other PU films showed DMPA-IPDI, polyester-IPDI and two different DMPA-IPDI-polyester hard domains. Finally, the adhesion properties of the PUDs and PU coatings were excellent and they were not influenced by the structural differences caused by adding DMPA in different stages of the synthesis.

## 1. Introduction

Environmental regulations of volatile organic compound (VOC) emissions in adhesives and coatings have been driving the fast growth of waterborne poly(urethane-urea) dispersions in the last decades [1,2,3,4]. Waterborne poly(urethane-urea) dispersions (PUDs) are colloidal systems consisting of hydrophobic poly(urethane-urea) (PU) particles dispersed in a continuous water phase. The stabilization of the particles is generally achieved by surface hydrophilic moieties of short internal emulsifier covalently bonded to the poly(urethane-urea) chains [5,6]. The first and most common internal emulsifier in the synthesis of PUDs is 2,2-bis(hydroxymethyl)propionic or dimethylolpropionic acid (DMPA) [6]. Typically, the synthesis of PUDs consists in three consecutive stages involving the formation of an isocyanate terminated urethane prepolymer, the chain extension with short amine and the dispersion of the poly(urethane-urea) in water. The structure of the PUD particles consists in soft and hard domains, and ionic interactions. The hard domains are produced by reacting isocyanate with DMPA, polyol and low molecular weight amine chain extender, and the soft segments are the polyol chains; the ionic interactions are due to the interactions between the carboxylic groups on the PUD particles and the quaternary ammonium cations in the water phase. The properties of the PUDs are related to their structure (i.e., structure-property relationship) which is determined by the raw materials, the nature and amount of internal emulsifier, the hard to soft segments ratio (NCO/OH ratio), and the polymerization conditions, among other [7,8,9,10,11,12,13,14,15,16,17].

The influence of the internal emulsifier amount, i.e., DMPA content, on the structure–properties relationship of PUDs has been widely studied [18,19,20,21,22]. In general, the mean particle size of the PUD decreased by increasing the amount of DMPA, because the increase of the ionic groups content and the interactions in the double electric layer of the particles [19,23,24,25,26,27,28,29,30,31,32,33]. Furthermore, the mean particle sizes and particle size distributions of the PUDs have been related to water absorption [5] and dispersion viscosity [15,23,25,26,34,35,36,37,38,39,40,41]. However, the structure–properties relationship in PUDs made by adding DMPA in different stages of their syntheses has been scarcely studied.

DMPA contains two hydroxyls and one carboxylic group. During PUD synthesis, the two hydroxyls groups react with the di-isocyanate during prepolymer formation producing hard segments. DMPA can be added before, during or after prepolymer formation, which will produce PUDs of different structure. Some previous studies have considered the structure and properties of PUDs synthesized by adding DMPA in different stages during prepolymer formation. The most of these studies proposed the addition of DMPA *before* prepolymer formation, i.e., DMPA was mixed with the polyol and, later, the di-isocyanate was added [18,25,42,43,44,45], the reactive –OH groups of DMPA and polyol competed to react with the isocyanate groups, this caused the random distribution of the diisocyanate-DMPA hard segments in the prepolymer chains. It has been proposed that the mixing of the polyol and DMPA at 60–70 °C for 30 min allowed a uniform distribution of the diisocyanate-DMPA hard segments in the poly(urethane-urea) backbone [26,29,46]. Alternatively, the polyol and the diisocyanate may react, and DMPA and catalyst can be added *during* prepolymer formation [38,47,48,49]. Some authors have also proposed the addition of DMPA *after* prepolymer formation for better control of the structure of the PUD [15,50,51,52,53].

Despite previous studies have considered the addition of DMPA during PUDs synthesis, relatively few have compared the structure–properties relationship in PUDs obtained by adding DMPA before, during and after prepolymer formation [26,38]. Harjunalanen and Lathtinen [26] have synthetized PUDs by reacting polyester polyol, isophorone diisocyanate (IPDI) and DMPA, the prepolymer mixing method was used. They synthesized three PUDs by adding DMPA at different stages of prepolymer formation: (i) PUD synthesized by adding polyester polyol and DMPA at the same time before addition of IPDI, and (ii) PUDs synthesized by adding DMPA after reacting polyester polyol and IPDI for 1 and 2 h. They found that the stage of the addition of DMPA affected the particle size and colloidal stability of the PUDs, the average particle sizes of the PUDs synthesized by adding DMPA during and after prepolymer formation were higher than the one of the PUD made by adding DMPA before prepolymer formation. Furthermore, those authors found superior tensile strength and melting temperature of the soft domains in the PU made by adding DMPA before prepolymer formation. In another study, Guo et al. [38] synthetized PUDs by reacting IPDI, polycaprolactone diol (PCL) and DMPA, ethylenediamine chain extender was used, and DPMA was added in two different ways: (i) two steps process—PCL and IPDI were reacted and later DMPA was added; and (ii) one step process—IPDI, PCL and DMPA were added together. They found that the particle size of the PUD made with the two steps process was bigger and the viscosity was lower than those of the PUD made with one step process, and the tensile strength and elongation of the PU were higher when the two steps process was used.

The influence of the stage of addition of DMPA before, during and after prepolymer formation on the structure–properties relationship of PUDs and PUs have been scarcely studied, the viscoelastic and adhesion properties have not been considered. The stage of the synthesis at which DMPA is added should produce structural dissimilarities in the PUDs. Thus, the addition of DMPA *before* and *after* prepolymer formation (i.e., the reaction of the isocyanate and the polyol) should produce ordered DMPA-isocyanate hard segments as well as polyol-isocyanate hard segments, but the distribution of the hard segments will be different. However, the addition of DMPA *during* prepolymer formation will produce random DMPA-isocyanate and polyol-isocyanate hard segments. Thus, depending on the stage of the synthesis at which DMPA is added, the structure of the hard segments and the degree of micro-phase separation will be different, the properties should be different too. Surprisingly, the structure–properties relationship in PUDs synthesized similarly but by adding DMPA *before*, *during* and *after* prepolymer formation have not been considered yet in the existing literature, and there are no studies considering the influence on the viscoelastic and adhesion properties. Therefore, in this study, DMPA was added *before*, *during* and *after* prepolymer formation during the synthesis of PUDs using MEK (methyl ethyl ketone) prepolymer method, and, in order to assess their structure–properties relationships, their structural, thermal, rheological, viscoelastic and adhesion properties have been compared.

## 2. Materials and Methods

### 2.1. Materials

Isophorone diisocyanate (IPDI), 2,2-bis(hydroxymethyl)propionic or dimethylolpropionic acid (DMPA) internal emulsifier, trimethylamine (TEA) neutralization agent, dibutyltin dilaurate (DBTDL) catalyst, and hydrazine monohydrate (Hz, 50–60 wt% purity) chain extender were used without further purification, all have been supplied by Sigma Aldrich (Sigma Aldrich Co. LLC, St. Louis, MO, USA). The polyol used was polyadipate of 1,6-hexanediol of molecular weight 2000 Da (Hoopol S-105-55—supplied by Synthesia Technology, Barcelona, Spain), it was dried at 80 °C under reduced pressure (300 mbar) for 2 h. Methyl ethyl ketone (MEK) was supplied by Jaber (Jaber S.A., Almansa, Spain) and deionized water was used as dispersed phase.

### 2.2. Synthesis of the Waterborne Poly(urethane-urea) Dispersions (PUDs)

The PUDs were synthesized by using MEK prepolymer method, the prepolymer mass was 250 g, NCO/OH ratio of 1.5 (the –OH groups of DMPA was considered) and 5 wt% DMPA were used.

The syntheses of the PUDs were carried out in 1 L reactor flask provided with five connections. Teflon rod coupled to Heidolph stirrer RZR-2000 (Heidolph, Kelheim, Germany) was placed in the central connection, one side connection was coupled to dry nitrogen stream (40 mL/min N_2_), a thermometer was placed in another, and the remaining connection was used for adding the reactants. The temperature during the synthesis was controlled with a thermostated water bath Frigiterm 6,000,382 (Selecta, Niles, IL, USA).

#### 2.2.1. Synthesis of the PUD by Adding DMPA before Prepolymer Dispersion (PUD *before* Prepolymer)

The scheme of the synthesis of the PUD by adding DMPA before prepolymer formation is shown in Figure 1. IPDI (61 g) and 75 g MEK (30 wt% with respect to the prepolymer mass) were added into the reactor flask at 50 °C under stirring at 150 rpm. Once the mixture was homogenized (after 30 min), DMPA (12.8 g) was added under stirring at 150 rpm and 60 °C for 90 min. Then, the polyol (177 g) was added under stirring at 150 rpm and, 30 min later, the catalyst (0.1 wt% with respect to the polyol mass, 0.17 g) was added and the reaction continued at 75 °C for 90 min. The free NCO content in the prepolymer (2.97%) was determined by *n*-dibutylamine titration. TEA (9.6 g) was added under stirring at 150 rpm, and 30 min later the prepolymer was obtained. Afterwards, distilled water (372 g) at 8 °C was slowly added over the prepolymer under stirring at 1000 rpm with a Cowles 18,133 MIL-1 stirrer (Lleal S.A., Granollers, Spain), the stirring speed was gradually increased from 1000 to 4000 rpm. Once the mixture was homogeneous (10 min at 4000 rpm), Hz chain extender (5.3 g) was added at room temperature under stirring at 1500 rpm for 30 min. Finally, MEK distillation was carried out under stirring at 150 rpm and 50 °C for 3 h, the residual pressure was gradually decreased from 300 to 100 mbar.

#### 2.2.2. Synthesis of the PUD by Adding DMPA during Prepolymer Formation (PUD *during* Prepolymer)

The synthesis of the PUD by adding DMPA *during* prepolymer formation is shown in Figure 2. The polyol (177 g) and 75 g MEK were added into the reactor flask at 50 °C under stirring at 150 rpm. Once the mixture was homogenized, IPDI (61 g) was added at 60 °C under stirring at 150 rpm for 90 min. Then, 12.8 g DMPA was added under stirring at 150 rpm and, 30 min later, 0.17 g catalyst was added. The experimental free NCO content in the prepolymer (2.50%) was determined by *n*-dibutylamine titration. The following stages of the synthesis were similar to the ones used in the synthesis of the PUD made by adding DMPA *before* prepolymer formation, 9.6 g TEA, 372 g distilled water and 4.9 g Hz chain extender were used.

#### 2.2.3. Synthesis of the PUD by Adding DMPA after Prepolymer Formation (PUD *after* Prepolymer)

The synthesis of the PUD by adding DMPA *after* prepolymer formation is shown in Figure 3. The polyol (177 g) and 75 g MEK were added into the reactor flask at 50 °C under stirring at 150 rpm. Once the mixture was homogenized for 30 min, 61 g IPDI were added under stirring at 150 rpm and, 30 min later, 0.17 g catalyst was added, the reaction was produced under stirring at 150 rpm and 70 °C for 90 min. Then, 12.8 g DMPA was added and the reaction continues under stirring at 150 rpm and 70 °C for 90 min. The free NCO content in the prepolymer (2.32%) was determined by *n*-dibutylamine titration. The following stages of the synthesis were similar to the ones used in the synthesis of the PUD made by adding DMPA *before* prepolymer formation, 9.6 g TEA, 372 g distilled water and 4.6 g Hz chain extender were used.

### 2.3. Experimental Techniques

#### 2.3.1. Characterization of the Waterborne Poly(urethane-urea)s (PUDs)

Solids content—The solids contents of the PUDs were determined in a DBS 60-3 thermobalance (Kern & Sohn GmbH, Balingen, Germany) by heating 1 g PUD at 105 °C for 15 min followed by heating at 120 °C until constant mass was obtained. Three replicates were carried out and averaged.

pH measurement—The pH values of the PUDs were measured at 25 °C in a pHmeter HI 8418 (Oakton Instruments, Vernon Hills, IL, USA) equipped with silver reference electrode. The pH was calculated as the average of three experimental determinations.

Particle size distribution—The mean particle sizes and particle size distributions of the PUDs were determined in a Laser Scattering Particle Size Distribution Analyser Horiba LA-950A2 (Horiba ABX SAS, Kyoto, Japan). One droplet of PUD was diluted in 1–1.5 mL distilled water before measurement. Two replicates for each PUD were measured and averaged.

Viscosity—The viscosities of the PUDs were measured at 25 °C in a DHR-2 rheometer (TA Instruments, New Castle, DE, USA) using coaxial cylindrical geometry according to DIN 53,019 standard. Two replicates for each PUD were measured and averaged.

#### 2.3.2. Characterization of the Solid Poly(urethane-urea) (PU) Films

Thin solid PU films were made in a TQC Automatic Film Applicator (TQC Instruments, Capelle aan den Ijssel, Netherlands). 10 g PUD was placed in Teflon mould of dimensions 12 × 24 × 0.2 mm and spread at a rate of 20 mm/s, the water was left to evaporate at room temperature for 24 h. Then, the PU films were completely dried in oven at 50 °C for 8 h. All PU films have thicknesses of 160–190 μm.

Attenuated total reflectance Fourier transform infrared (ATR-IR) spectroscopy—The structure of the PU films was assessed in an Alpha spectrometer (Bruker Optik GmbH, Ettlingen, Germany) using a germanium prism. An incidence angle of the IR radiation of 45° was used, 64 scans were recorded and averaged with a resolution of 4 cm^−1^.

Differential scanning calorimetry (DSC)—The structure of the PU films was assessed in a DSC Q100 differential scanning calorimeter (TA Instruments, New Castle, DE, USA) under nitrogen atmosphere (flow rate = 100 mL/min). 8.5 mg PU film was placed in hermetically sealed aluminum pan and heated from −80 to 120 °C by using a heating rate of 10 °C/min. Afterwards, the PU film was cooled down to −80 °C at a cooling rate of 10 °C/min, and, then, a second DSC heating run from −80 to 250 °C was carried out by using a heating rate of 10 °C/min. The glass transition temperatures (T_g_s) and thermal events of the PU films were determined from the second DSC heating runs.

X-ray diffraction (XRD)—The crystallinity of the PU films was determined in a Bruker D8-Advance diffractometer (Bruker, Ettlingen, Germany), the wavelength of copper Kα radiation (1.540598 A), copper cathode, and nickel filter with Göbel mirror were used. A scanning of 2θ angles between 5° and 90° in 0.05° steps acquired at 3 s/step was carried out.

Thermal gravimetric analysis (TGA)—The structure and thermal properties of the PU films were assessed in a TGA Q500 equipment (TA Instruments, New Castle, DE, USA). 10 mg PU film were placed in platinum crucible and heated under nitrogen (flow rate: 100 mL/min) from room temperature up to 800 °C, by using a heating rate of 10 °C/min.

Plate–plate rheology—The rheological properties of the PU films were determined in a Discovery HR-2 hybrid rheometer (TA Instruments, New Castle, DE, USA) using plate–plate geometry. Temperature sweep experiments in the region of linear viscoelasticity were carried out and the variations of the storage (G’) and loss (G’’) moduli as a function of the temperature were recorded; the gap was 0.4 mm and the frequency was 1 Hz.

Dynamic mechanical thermal analysis (DMA)—Rectangular PU films of dimensions 18 mm × 11 mm × 2 mm were prepared in silicone moulds filled with PUDs which were left to water evaporation at 25 °C for 24 h followed by heating at 50 °C for 4 h. The viscoelastic properties of the PU films were determined in a Q800 DMA equipment (TA Instruments, New Castle, DE, USA) in single cantilever geometry. Rectangular samples were mounted vertically in the clamps and a frequency of 1 Hz was selected. The temperature was varied from −100 to 200 °C, a heating rate of 5 °C/min was used; the preload force was 0.01 N and the force track was set to 125%. The amplitude of the vertical oscillation was 15 µm.

#### 2.3.3. Adhesion Properties

Cross-hatch adhesion test—The adhesion of the PU coatings obtained by applying PUDs over 6 cm × 6 cm 304 AISI stainless steel plates was evaluated by cross-hatch adhesion test following ASTM D3359-02 standard. About 1 g PUD was spread on isopropanol wiped stainless steel plate by means of a metering rod of 200 µm for controlling coating thickness. Once the water was removed completely at 25 °C for 24 h followed by heating at 50 °C for 4 h, six parallel cuts was made on the PU coating by means of a special cutter for obtaining a right-angle lattice pattern and, then, a standard Tesa^®^ tape was applied on the cut pattern and removed by peeling. The number of PU coating squares removed by the tape was assessed by using a magnifier, and the cross-hatch adhesion was determined according to ASTM D3359-02 scale given in Table 1. Three replicates were carried out and averaged for each PU coating.

T-peel strength test—The adhesion properties of the PUDs were determined by T-peel tests of plasticized poly(vinyl chloride (PVC)/PUD/plasticized PVC joints. The plasticized PVC test samples had dimensions of 30 mm × 150 mm × 5 mm, and they were methyl ethyl ketone wiped for plasticizer removal, allowing the solvent to evaporate for 30 min under open air. Then, 1 g PUD was applied by brush to each PVC strip to be joined and, after water evaporation at 25 °C (it took about 90 min), the solid adhesive films were heated suddenly at 80 °C for 10 s under infrared radiation (reactivation process). The PVC strips were immediately placed in contact and a pressure of 0.8 MPa was applied for 10 s to achieve a suitable joint. The T-peel strength was measured 1 and 72 h after joint formation in an Instron 4411 universal testing machine (Instron Ltd., Buckinghamshire, UK) by using a crosshead speed of 100 mm/min (Figure 4). The values obtained were the average of five replicates.

## 3. Results and Discussion

### 3.1. Characterizaction of the PUDs

Different PUDs were synthesized by adding DMPA before (PUD *before* prepolymer), during (PUD *during* prepolymer) and after (PUD *after* prepolymer) prepolymer formation. The addition of DMPA *before* and *after* prepolymer formation (i.e., the reaction of the isocyanate and the polyol) should produce ordered DMPA-isocyanate hard segments as well as polyol-isocyanate hard segments, but the distribution of the hard segments will be different. However, the addition of DMPA *during* prepolymer formation will produce random DMPA-isocyanate and polyol-isocyanate hard segments. The pH values of the PUDs are basic (8.0–9.2) and typical of PUDs intended for adhesives and coatings [54], the pH value of PUD *after* prepolymer is less basic than in the others likely due to lower amount of quaternary ammonium anions in the PUD, this may be ascribed to longer polymerization time, larger molecular weight of the poly(urethane urea) chains and more difficult breakup during the dispersion with water.

All PUDs are stable for at least 6 months after synthesis and they are bluish translucent (Figure 5). The solids contents of the PUDs are 38–40 wt% (Table 2), they are close to the targeted 40 wt% value. The degree of transparency and color of PUDs can be related to their mean particle size, particle size distribution and method of synthesis. MEK prepolymer method generally provides PUD with a bluish appearance. The three PUDs of Figure 5 are somewhat similar because their relatively low solids content (they have 60% water at least) and the existence of mono-modal particle size distributions [55].

The mean particle sizes of the PUDs are given in Table 2. The particle size of PUD *after* prepolymer is significantly higher than the ones of the other PUDs, in agreement with previous studies [26,38]. The lowest particle sizes correspond to the PUDs in which DMPA was added *before* and *during* prepolymer formation. When DMPA is added before prepolymer formation, all DMPA reacts with IPDI and the remaining NCO groups reacts with the polyol, so the molecular weight of the prepolymer will be smaller than the one of the PUD synthesized by adding DMPA *after* prepolymer formation, the particle size should be higher in PUD *after* prepolymer. When DMPA is added during prepolymer formation, both DMPA and polyol compete with IPDI to react and the molecular weight of the prepolymer should be intermediate. On the other hand, the larger particle size of PUD *after* prepolymer should impart higher hydrophobicity than in PUD *before* prepolymer because of more carboxylic groups on the surface of the particles. Furthermore, all PUDs show relatively ample mono-modal particle size distributions, so they were curve fitted for determining the main particle sizes. According to Figure 6, all PUDs show the existence of two main particle sizes which are closer in PUD *before* and *during* prepolymer, but they differ more in PUD *after* prepolymer, this can be ascribed to longer polymerization time.

Figure 7 shows the variation of the viscosity of the PUDs at 25 °C as a function of the shear rate, the viscosities measured at 10^3^ s^−1^ shear rate are given in Table 2. The viscosity of PUD *during* prepolymer is higher than in the other PUDs, and shear thinning is exhibited (PUD *before* prepolymer and PUD *after* prepolymer show Newtonian rheological behavior). Shear thinning can be related to stronger interactions between the surfaces of the particles in PUD *during* prepolymer, i.e., higher concentration of carboxylic groups on the particles surface is obtained in PUD *during* prepolymer.

It has been demonstrated that there is not a simple relationship between the molecular weight, the particle size and the viscosity of PUDs. The particle size of the PUDs is mainly determined by the hydrophilicity and the hydrodynamic volume of the particles, this is caused by the swelling when water is added during PUD synthesis [56]. Therefore, depending on the hydrophilicity and the hydrodynamic volume of the particles, different trends in particle size and viscosity of PUDs can be obtained. The degree of hydrophilicity is somewhat similar in all PUDs because the same amount of DMPA is added but in PUD *after* prepolymer, the molecular weight of the poly(urethane urea) chains is higher than in the other, this lead to increased hydrophobicity and, therefore, higher mean particle size. At the same time, the large molecular weight of the poly(urethane urea) chains in PUD after prepolymer makes breakup difficult during the dispersion with water and the viscosity decreases. On the other hand, Guo et al. [38] found similar trend, i.e., the lowest viscosity corresponded to PUD with higher mean particle size.

### 3.2. Characterizaction of the PU Films

The chemical structure of the PU films was studied by ATR-IR spectroscopy. All PU films show relatively similar ATR-IR spectra (Figure 8) in which the bands of the soft segments correspond to C–H stretching of aliphatic chains at 2920 and 2860 cm^−1^, asymmetric and symmetric −CH_2_ bending at 1460 and 1402 cm^−1^ respectively, and the characteristic stretching bands of C–O and C–O–C groups of the polyester polyol are located at 1260, 1170 and 1034 cm^−1^. The bands of the hard segments correspond to N–H stretching at 3330 cm^−1^, C–N stretching of urea groups at 1520 cm^−1^, and C=O stretching of urethane, urea and free ester groups of the polyol at 1730 cm^−1^. The wavenumber at which the different bands appear are somewhat similar in all PU films, irrespective of the stage at which DMPA is added during the synthesis.

The relative percentages of the different C=O species in the PU films were determined from the curve fitting of the carbonyl region (1800–1600 cm^−1^) of the ATR-IR spectra. Figure 9 shows, as typical example, the curve fitting of the carbonyl region of PU *during* prepolymer film. The curve fitting displays four kind of carbonyl species: free ester and free urethane groups at 1730 cm^−1^, associated by hydrogen bond (H-bonded) urethane groups at 1722–1720 cm^−1^, free urea groups at 1707–1705 cm^−1^, and associated by hydrogen bond (H-bonded) urea groups at 1688–1683 cm^−1^ [2]. According to Table 3, similar percentages of all kind of C=O species are observed in all PU films, irrespective of the stage at which DMPA is added during the synthesis, and similar percentages of urethane and urea groups are found.

The structure of the PU films was assessed by DSC. Figure 10a shows the DSC traces of the second heating runs of the PU films which exhibit the glass transition of the soft segments at low temperature (−57 to −69 °C), and the cold crystallization (−8 to 4 °C) and melting (45 °C) of the soft segments. The glass transition temperature is lower and the melting enthalpy is higher in PU *before* prepolymer than in the rest of the PU films. Whereas the glass transition temperatures of PU *during* prepolymer and PU *after* prepolymer are somewhat similar, the cold crystallization appears at lower temperature and with higher enthalpy in PU *during* prepolymer. During the DSC cooling run, the soft segments crystallize poorly at 29 °C in PU *after* prepolymer (Figure 10b) but marked crystallization at −0.2 °C is produced in PU *before* prepolymer. Therefore, the degree of micro-phase separation in the PU films depends on the stage at which DMPA is added during the synthesis, the higher degree of micro-phase separation corresponds to PU *before* prepolymer. On the other hand, more net interactions between the soft domains is found in PU *before* prepolymer and less net in PU *after* prepolymer.

The crystallinity of the PU films was also determined by X-ray diffraction. The X-ray diffractograms are very similar in all PU films (Figure 11) and the most intense diffractions are produced at 2θ values of 21.3° and 23.9°, they are due to the crystallites of soft domains; however, the intensities of these diffraction peaks are lower in PU *after* prepolymer, in agreement with its lower crystallization enthalpy evidenced by DSC. On the other hand, all X-ray diffractograms show a broad peak at 2θ values between 12° and 25° due to amorphous structure which is more marked in PU *during* prepolymer and less important in PU *before* prepolymer, this agrees with the trend in the crystallization of the soft domains evidenced by DSC experiments.

The thermal stability of the PU films was assessed by TGA. Figure 12 shows that the TGA curves of PU *before* prepolymer and PU *after* prepolymer are similar, the thermal stability of PU *during* prepolymer is lower, this is also evidenced by the lower temperatures at which 5 and 50% mass are lost (Table 4).

The structure of the PU films can also be assessed from the thermal decompositions evidenced in the DTGA (derivative of the weight) curves [7]. Figure 13 shows different thermal decompositions in the PU films, the temperatures at which they are produced differ depending on the stage at which DMPA is added during synthesis (Table 5). The small degradation at 52 °C is due to residual moisture and appears in all PU films. The thermal degradations of the urethane hard domains appear between 163 and 288 °C, the ones of the urea hard domains can be distinguished at 341–348 °C, and the thermal degradations of the soft domains are found at 393–433 °C. The structures of the PU films are dominated by the soft domains that account 66–70 wt%, they are similar in all PU films, irrespective of the stage at which DMPA is added. The main structural differences in the PU films derived from the hard domains. PU *during* prepolymer shows the thermal decompositions of DMPA-IPDI hard domains at 183 °C, urethane hard domains at 283 °C and urea hard domains at 341 °C. However, the other PU films have more complex structure of the hard domains. Thus, PU *before* prepolymer shows the thermal decompositions of DMPA-IPDI hard domains at 196 °C and urea hard domains at 348 °C, but several urethane hard domains at 237, 262 and 286 °C appear, this indicates the existence of different DMPA-IPDI-polyester hard domains. In fact, the total weight loss of the urethane hard domains is 12 wt% and the one of the urea hard domains is 19 wt% because the shorter prepolymer produced during PUD synthesis (Figure 1), the formation of higher amount of urea hard domains is favoured. The structure of the hard domains in PU *after* prepolymer is also complex and shows the thermal decomposition of DMPA-IPDI and polyester-IPDI hard domains at 163 and 197 °C, and urea hard domains at 345 °C, but several urethane hard domains at 230 and 288 °C appear, this indicates the existence of two different DMPA-IPDI-polyester hard domains. Furthermore, the total weight loss of the urethane hard domains is 14 wt% and the one of the urea hard domains is 15 wt% because the longer prepolymer produced during PUD synthesis (Figure 3), the formation of less urea hard domains than in PUD *before* prepolymer is produced. Therefore, the micro-phase separation differs in the PU films depending on the stage at which DMPA is added during synthesis.

The viscoelastic properties of the PU films were studied by dynamic mechanical analysis (DMA). Figure 14a shows the variation of the storage modulus (E’) of the PU films as a function of the temperature and the glassy, glass transition, rubbery plateau and melting of the soft segments regions are noticed. The lowest storage modulus and early melting of the soft segments correspond to PU before prepolymer because the shorter prepolymer produced during synthesis (Figure 1) and the highest storage modulus and later melting of the soft segments correspond to PU after prepolymer because the longer prepolymer produced during its synthesis (Figure 3). The structural relaxations in the PU films are noticed in Figure 14b, the PU films show the α and β relaxations. The maximum tan delta values correspond to PU during prepolymer (Table 6) because less complex hard domains structure, and the lowest T_β_ corresponds to PU after prepolymer.

The rheological properties of the PU films were evaluated by plate–plate rheology. Figure 15 show the variation of the storage modulus (G’) and loss modulus (G’’) as a function of the temperature for PU films, and they show a cross at 68–81 °C, i.e., below this temperature the elastic rheological regime is dominant and above 68–81 °C the viscous rheological regime is dominant. The temperatures and moduli at the cross of G’ and G’’ are somewhat similar in PU before prepolymer and PU during prepolymer, but the temperature is higher and the modulus is lower in PU after prepolymer. Therefore, PU after prepolymer shows lower degree of micro-phase separation than the other PU films.

### 3.3. Adhesion Properties of the PUDs

The adhesion properties of the PUDs were determined by T-peel tests of plasticized PVC/PUD/plasticized PVC joints and the adhesion of the PU coatings on stainless steel 304 plate was assessed by cross-adhesion tests.

The T-peel strength values of plasticized PVC/PUD/plasticized PVC joints are given in Table 7. The T-peel strengths are similar in all joints, irrespective of the stage at which DMPA is added during PUD synthesis. When the T-peel strength is measured 1 h after joint formation, some water remains in the PU film, this justify the lower peel strength and the cohesive failure of the adhesive during peel test. However, 72 h after joint formation, the water in the PU film is completely removed and, therefore, higher peel strength is obtained, and the loci of failure of the joints change to cohesive rupture of the substrate. The structural differences of the PUDs due to the addition of DMPA at different stage of the synthesis do not determine their T-peel adhesion properties.

The cross-hatch adhesion of PU coatings on stainless steel plate (Figure 16) was evaluated according to ASTM D3359-09 standard, and the values obtained are given in Table 7. As for the T-peel strength values, the cross-hatch adhesion values of the PU coatings are excellent and similar in all PU films.

## 4. Conclusions

The mean particle sizes of the PUD made by adding DMPA *after* prepolymer formation was higher than for the rest, all PUDs showed relatively ample mono-modal particle size distributions, and the viscosity of the PUD synthesized by adding DMPA *during* prepolymer formation was higher and showed shear thinning.

The structure of all PU films shows similar percentages of free and associated by hydrogen bond urethane groups, and free urea and associated by hydrogen bond urea groups, irrespective of the stage at which DMPA was added during the synthesis. All PU films exhibited the glass transition of the soft segments at low temperature, a cold crystallization and the melting of the soft segments were also found. The glass transition temperature was lower, the crystallization of the soft segment was noticeable and the melting enthalpy was higher in the PU synthesized by adding DMPA *before* prepolymer formation than in the rest of the PU films; however, the cold crystallization appeared at lower temperature and with higher enthalpy, and the melting enthalpy of the soft segments was higher in the PU synthesized by *adding* DMPA during prepolymer formation. Therefore, the degree of micro-phase separation in the PU films depended on the stage at which DMPA was added during synthesis, the higher degree of micro-phase separation was found in the PU synthesized by adding DMPA *before* prepolymer formation. On the other hand, the interactions between the soft domains are more net in PU *before* prepolymer and less net in PU *after* prepolymer.

The thermal stability of the PU synthesized by adding DMPA *during* prepolymer formation was lower than in the other PU films, and their structures were dominated by the soft domains, they were similar in all PU films, irrespective of the stage at which DMPA is added. The main structural differences in the PU films derived from the hard domains. The PU synthesized by adding DMPA *during* prepolymer formation showed DMPA-IPDI, urethane and urea hard domains, whereas the other PU films had more complex structure of the hard domains made of several DMPA-IPDI, polyester-IPDI and DMPA-IPDI-polyester hard domains.

The lowest storage modulus and the early melting corresponded to the PU synthesized by adding DMPA *before* prepolymer formation because shorter prepolymer was produced during synthesis, and the highest storage modulus and later melting corresponded to the PU synthesized by adding DMPA *after* prepolymer formation because longer prepolymer was formed. All PU films showed a cross of G’ and G’’ and the temperatures at the cross was higher and the modulus was lower in the PU synthesized by adding DMPA *after* prepolymer formation because its lower degree of micro-phase separation.

The adhesion properties of the PUDs and PU coatings were excellent and they were not determined by the structural differences caused by adding DMPA during different stages of the synthesis.

## Figures and Tables

**Figure 1 polymers-12-02478-f001:**
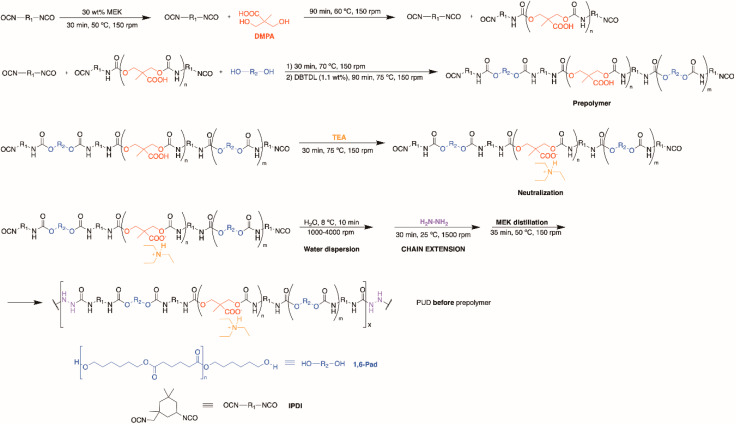
Scheme of the synthesis of the poly(urethane-urea) (PUD) by adding dimethylolpropionic acid (DMPA) *before* prepolymer formation.

**Figure 2 polymers-12-02478-f002:**
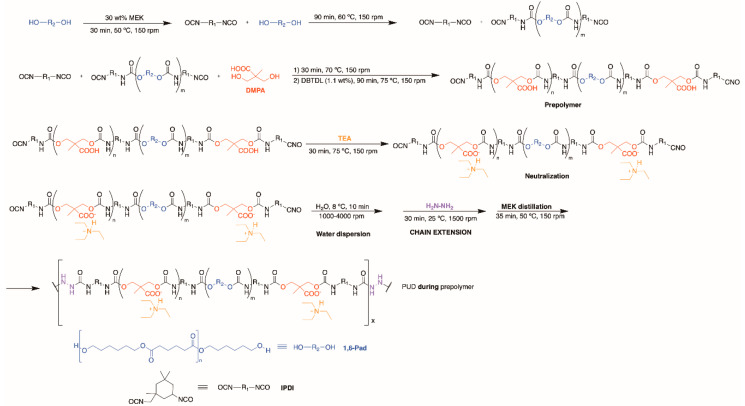
Synthesis of the PUD by adding DMPA *during* prepolymer formation.

**Figure 3 polymers-12-02478-f003:**
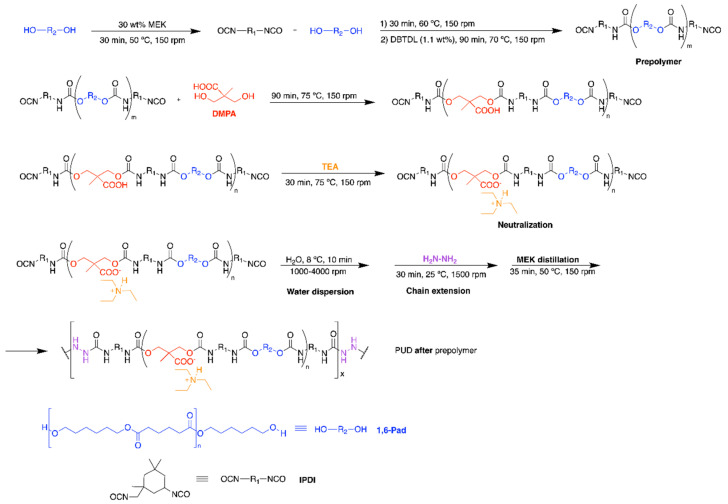
Synthesis of the PUD by adding DMPA *after* prepolymer formation.

**Figure 4 polymers-12-02478-f004:**
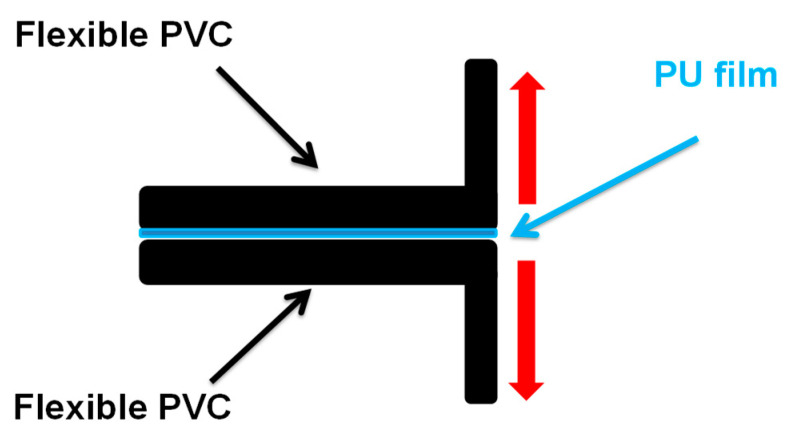
Scheme of T-peel test of plasticized poly(vinyl chloride (PVC)/PUD/plasticized PVC joints.

**Figure 5 polymers-12-02478-f005:**
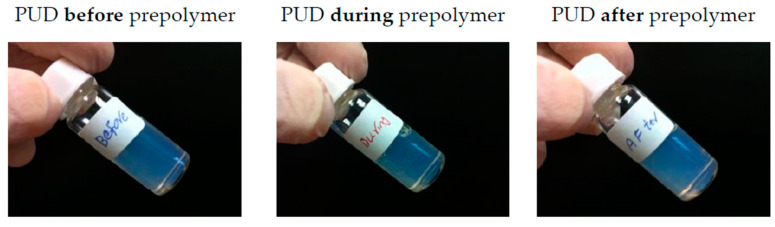
Appearance of the PUDs.

**Figure 6 polymers-12-02478-f006:**
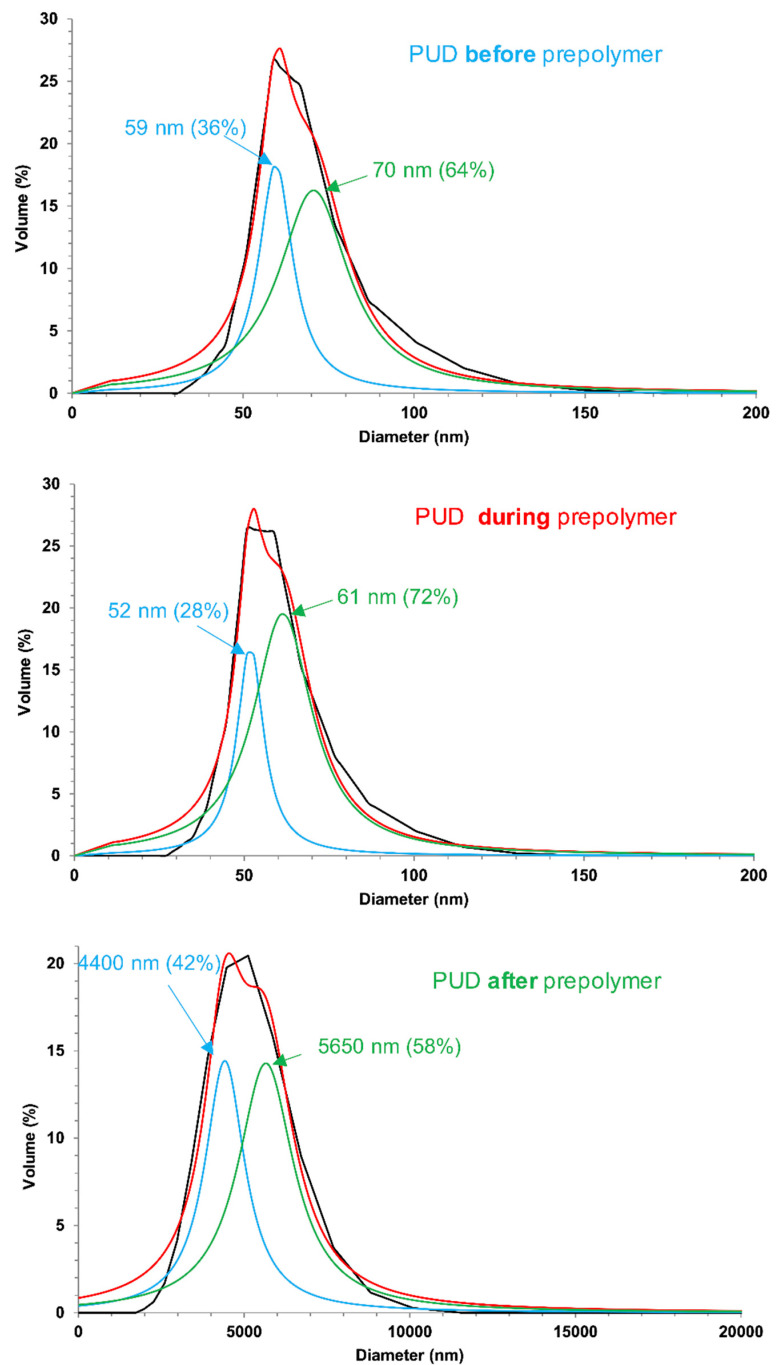
Curve fitting of the particle size distribution curves of the PUDs. Experimental data correspond to the black curves and the curve fitted data correspond to the red curves.

**Figure 7 polymers-12-02478-f007:**
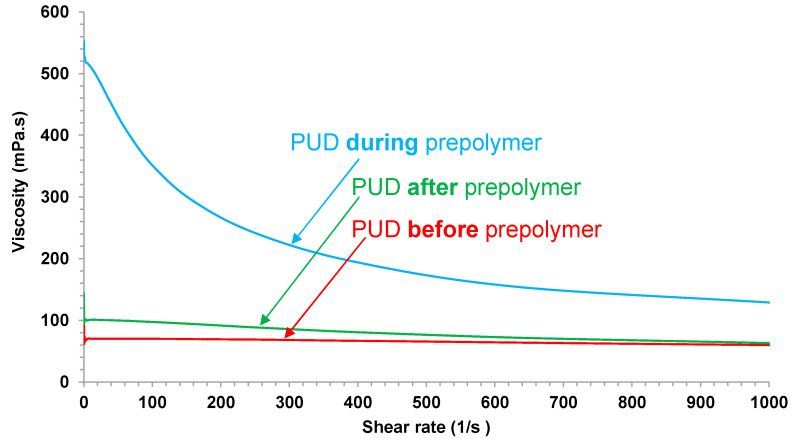
Variation of the viscosity at 25 °C of the PUDs as a function of the shear rate. Coaxial cylindrical geometry experiments.

**Figure 8 polymers-12-02478-f008:**
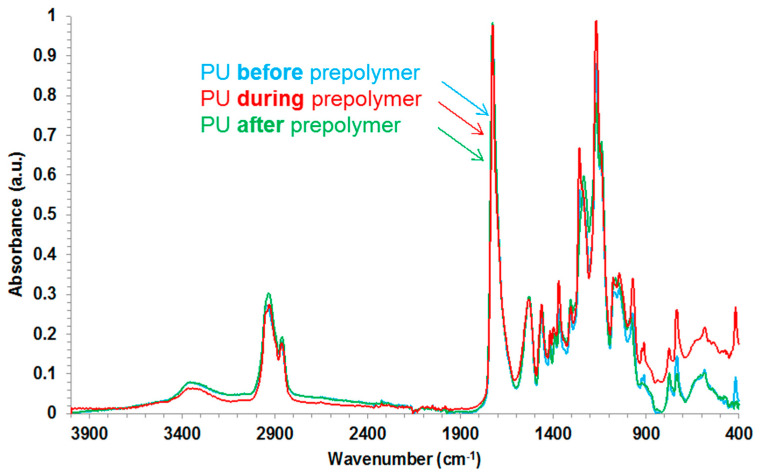
ATR-IR spectra of the PU films.

**Figure 9 polymers-12-02478-f009:**
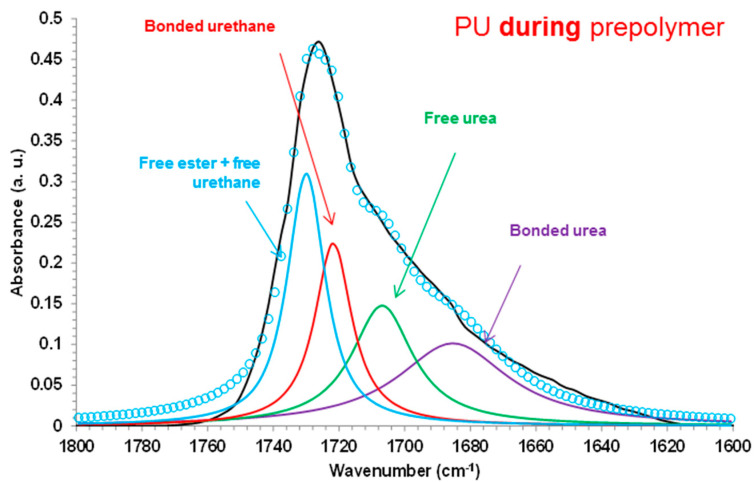
Curve fitting of the carbonyl region (1800–1600 cm^−1^) of the ATR-IR spectrum of PU during prepolymer film.

**Figure 10 polymers-12-02478-f010:**
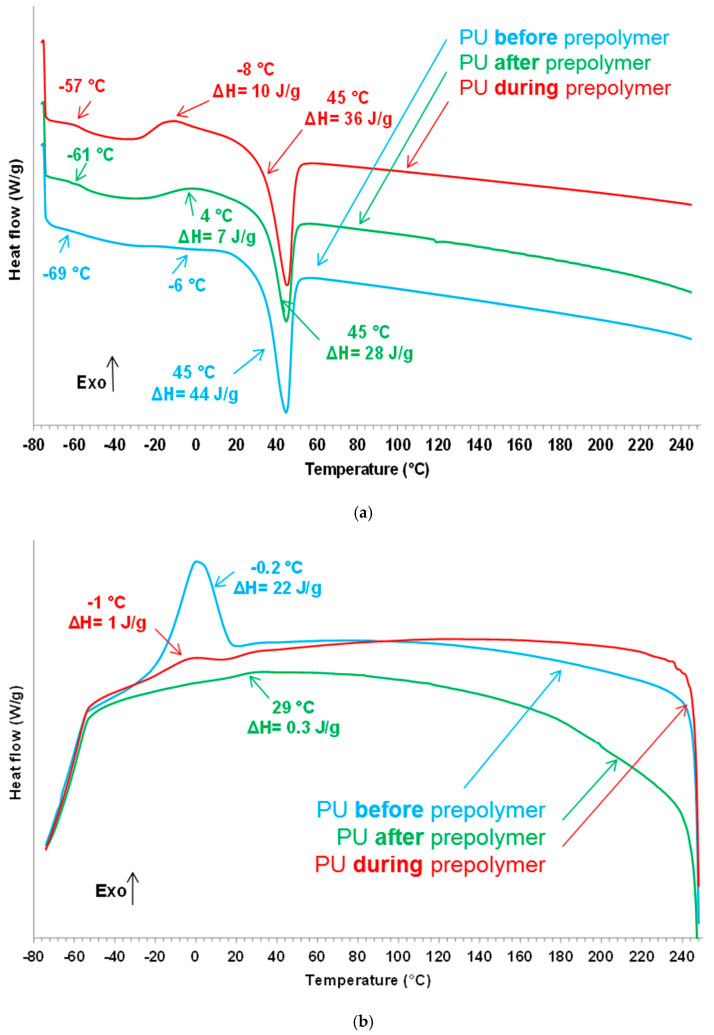
Differential scanning calorimetry (DSC) traces of the PU films: (**a**) second DSC heating run; (**b**) DSC cooling run.

**Figure 11 polymers-12-02478-f011:**
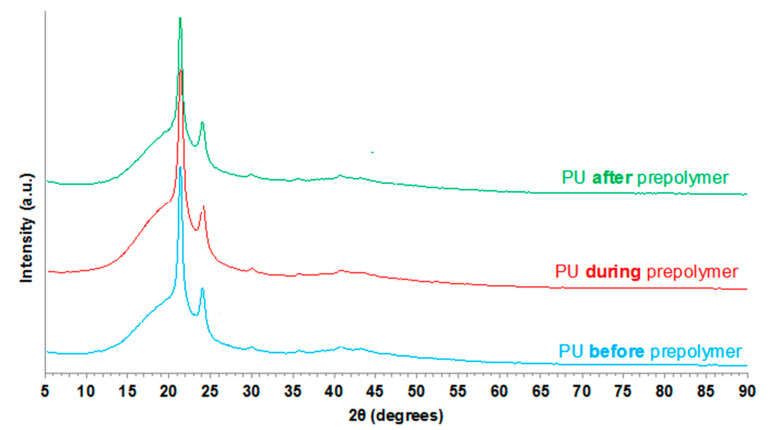
X-ray diffractograms of the PU films.

**Figure 12 polymers-12-02478-f012:**
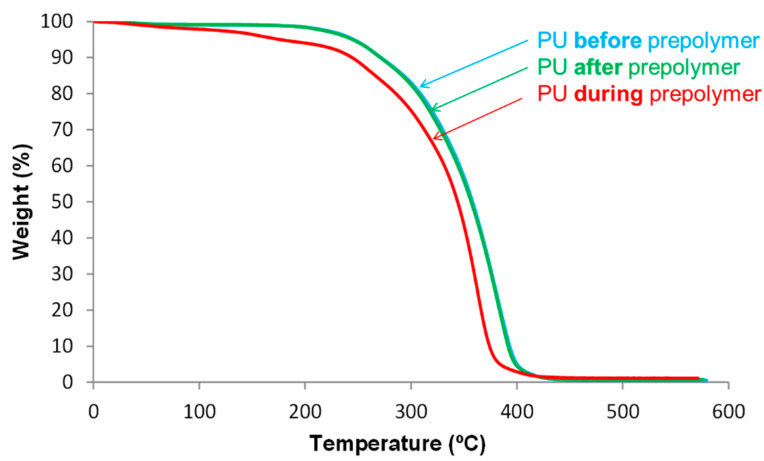
Variation of the weight as a function of the temperature for PU films. TGA experiments.

**Figure 13 polymers-12-02478-f013:**
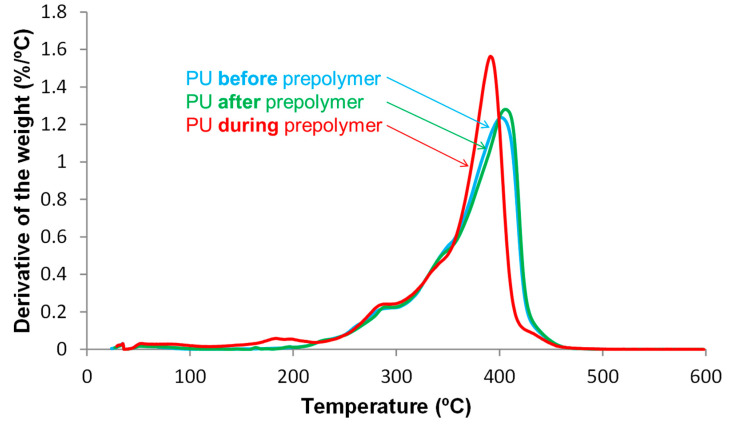
Variation of the derivative of the weight as a function of the temperature for PU films. DTGA experiments.

**Figure 14 polymers-12-02478-f014:**
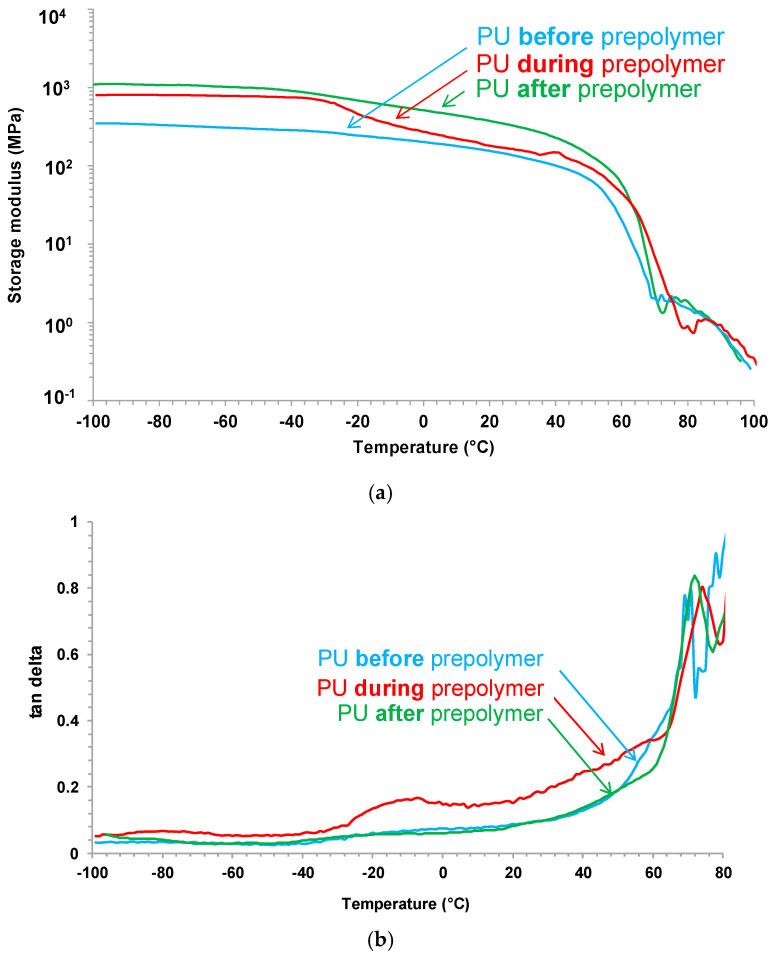
(**a**) Variation of the storage modulus (E’) as a function of the temperature, and (**b**) variation of tan delta as a function of the temperature for PU films. DMA experiments.

**Figure 15 polymers-12-02478-f015:**
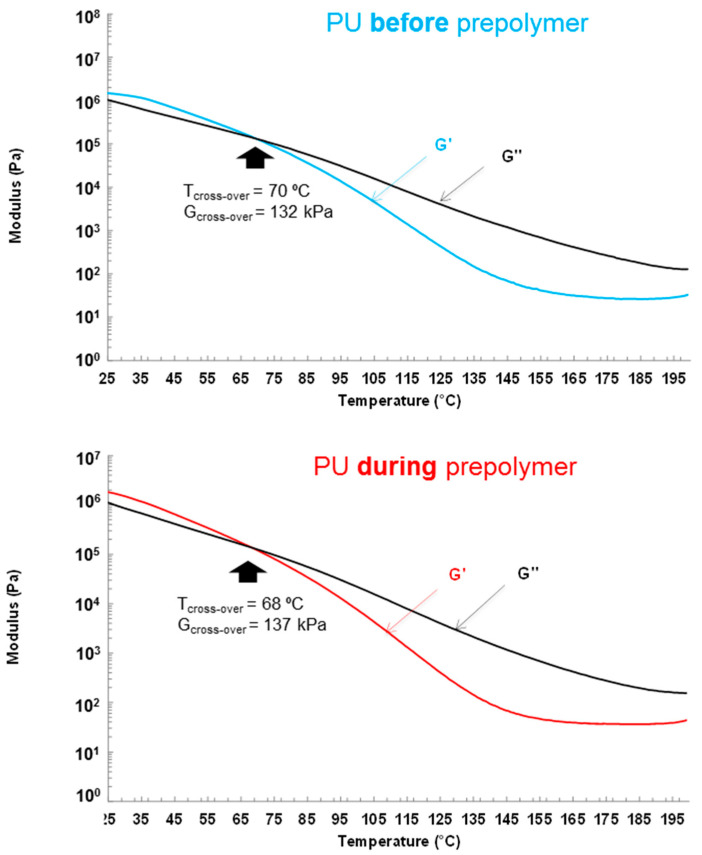
Variation of the storage (G’) and loss (G’’) moduli as a function of the temperature for PU films. Plate–plate rheology experiments.

**Figure 16 polymers-12-02478-f016:**
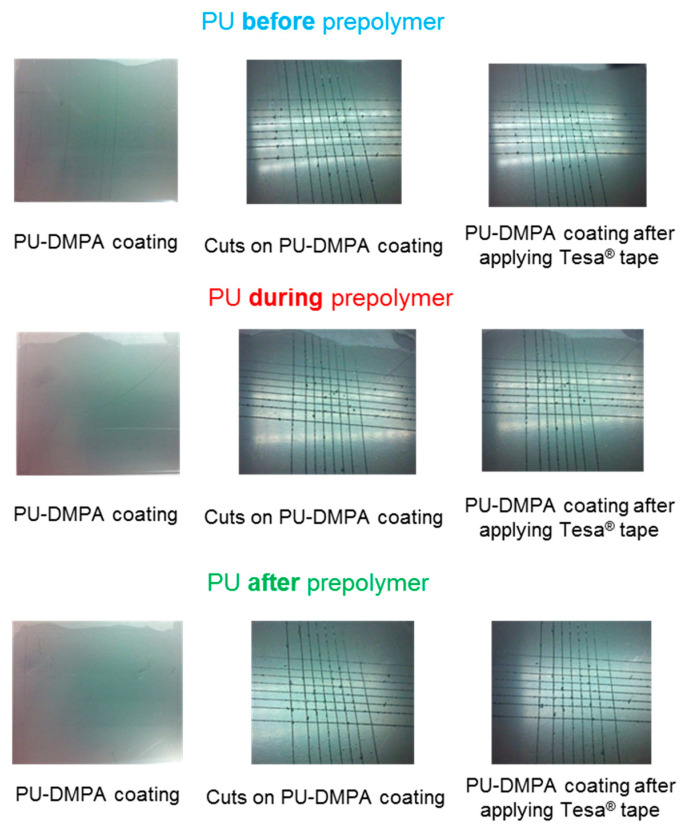
Appearance of the PU coatings synthesized by adding DMPA at different stages during synthesis on stainless steel plate. ASTM D3359-09 standard.

**Table 1 polymers-12-02478-t001:** Classification of cross-hatch adhesion values according to ASTM D3359-09 standard.

Classification	Area Removed (%)	Surface of Cross-Cut Area from Each Flaking Produced by Six Parallel Cuts
5B	0 (none)	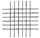
4B	Less than 5	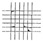
3B	5–15	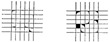
2B	15–35	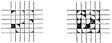
1B	35–65	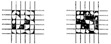
0B	Greater than 65	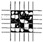

**Table 2 polymers-12-02478-t002:** Solids content, pH, mean particle size and viscosity values taken at a shear rate of 10^3^ s^−1^ measured at 25 °C of the PUDs.

PUD	Solids Content (wt%) ^a^	pH	Mean Particle Size (nm)	Viscosity at 10^3^ s^−1^ (mPa·s)
PUD *before* prepolymer	39.0 ± 0.1	8.9 ± 0.0	60	79
PUD *during* prepolymer	39.6 ± 0.8	9.2 ± 0.3	52	128
PUD *after* prepolymer	37.5 ± 0.6	8.0 ± 0.0	4485	63

^a^ Targeted solids content was 40 wt%.

**Table 3 polymers-12-02478-t003:** Contributions of the different carbonyl species of the PU films. Curve fitting of the carbonyl region of the ATR-IR spectra.

Wavenumber (cm^−1^)	Relative Contribution of Species (%)		
	PU *before* Prepolymer	PU *during* Prepolymer	PU *after* Prepolymer
1730 (Free ester + free urethane)	28	28	28
1722–1720 (H-bonded urethane)	21	21	22
1707–1705 (Free urea)	23	26	22
1688–1683 (H-bonded urea)	28	25	27

**Table 4 polymers-12-02478-t004:** Temperatures at which 5 (T_5%_) and 50 (T_50%_)% mass are lost in the PU films. TGA experiments.

PU	T_5%_ (°C)	T_50%_ (°C)
PU *before* prepolymer	267	379
PU *during* prepolymer	205	334
PU *after* prepolymer	269	381

**Table 5 polymers-12-02478-t005:** Temperatures and weight losses of the thermal decompositions of the PU films. DTGA experiments.

PU Film	PU *before* Prepolymer		PU *during* Prepolymer		PU *after* Prepolymer	
Thermal Decomposition	T (°C)	Weight Loss (%)	T (°C)	Weight Loss (%)	T (°C)	Weight Loss (%)
Degradation 1	52	1	52	1	52	1
Degradation 2	196	1	183	4	163,197	2
Degradation 3	237,262	4	-	-	230	2
Degradation 4	286	7	283	16	288	10
Degradation 5	348	19	341	12	345	15
Degradation 6	403	68	393,433	66	408	70

**Table 6 polymers-12-02478-t006:** Values of the temperature and tan delta of the relaxations of the PU films. DMA experiments.

PU Film	T_β_ (°C)	Tan Delta_β_	T_α_ (°C)	Tan Delta_α_
PU *before* prepolymer	−82	0.02	−12	0.03
PU *during* prepolymer	−80	0.08	−8	0.19
PU *after* prepolymer	−90	0.02	−12	0.03

**Table 7 polymers-12-02478-t007:** T-peel strength values of plasticized PVC/PUD/plasticized PVC joints and cross-hatch adhesion values evaluated according to ASTM D3359-09 standard.

PUD	T-Peel Strength ^a^		Cross-Hatch Adhesion (ASTM D3359-09)
	After 1 h	After 72 h	
PU *before* prepolymer	5.8 ± 0.7 (CA)	13.0 ± 3.0 (CS)	5B
PU *during* prepolymer	5.6 ± 0.1 (CA)	13.0 ± 2.0 (CS)	5B
PU *after* prepolymer	6.5 ± 0.5 (CA)	11.0 ± 1.0 (CS)	5B

^a^ Locus of failure: Cohesive failure of the adhesive (CA), Cohesive failure of the substrate (CS).

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
