# Peer review of "Structure–Properties Relationship in Waterborne Poly(Urethane-Urea)s Synthesized with Dimethylolpropionic Acid (DMPA) Internal Emulsifier Added before, during and after Prepolymer Formation"

_polymers, 2020, doi:10.3390/polym12112478_

Round 1

Reviewer 1 Report

1. Please check a spelling mistake on page 3 line 100.

2. Can the authors explicitly describe any significant change in PUDs because of employing DMPA in this study? It is hard to find/see any reasonable match between the purpose of the study and the results obtained from the investigations.

3. From Fig. 5 what should be understood/realized by a careful reader on the appearance of the materials? Are they looked like the same or different and then "why"?

4. In Table 2 (page 8) presented data exhibits that in the case of "after pre-polymerization" PUDs particle size increases by roughly a factor of 75 and 80 compared to before and during prepolymers. Does not this result indicate the mass of the after polymerization product increased? If so, why did the viscosity reached a minimum?

5. Change in adhesion properties, in before-during-after prepolymers cannot be speculated from the data presented in Table 7. Can the authors exhibit any reasonable change of adhesion properties in the resultant materials compared to the initial one?

Reviewer 2 Report

The authors present a comprehensive study of the DMPA addition methods' influence on the structure and properties of PUD. In this study, DMPA was added before, during, and after prepolymer dispersion synthesis.
PUDs were characterized by pH, viscosity, and particle size measurements, and the structure of the polyurethane-urea (PU) films was assessed by FTIR, DSC, X-ray, TGA, plate-plate rheology, and dynamic mechanical thermal analysis. More importantly, since PUDs are mainly used in coating and adhesive industries, the adhesion properties of the PUDs were measured by cross-hatch adhesion and T-peel test. The findings of the influence of DPMA addition sequence provide a good reference and guidance for PUDs manufacturing.
The reviewer finds that the authors have presented a solid work and written the manuscript with clarity and command. The synthesis and following materials characterizations are the reviewer finds thorough, providing data that unambiguously supported the arguments in the manuscript. Overall, the reviewer commends the quality of work and presentation.

Reviewer 3 Report

In this work by Fuensanta et al., polyurethane dispersion (PUD) was prepared by using DMPA as an internal emulsifier that was added before, during and after prepolymer preparation. The colloidal PUDs and films were characterized with their structure-property relationship. The work has been done with good sequence and the material was characterized well. This manuscript is suitable for the publication in Polymers after considering few minor comments.

  1. The Authors should include a discussion on structural dissimilarities in PU synthesized in 3 different processes. For example, Figure 2, adding DMPA during prepolymer formation might give you the prepolymer with random distribution of DMPA.
  2. Figure 6, what is the significance of bimodal particle size distribution. Also, please mention the y-axis values are whether intensity/volume/number %.

Round 2

Reviewer 1 Report

I do not see any novelty in the work or significant/acceptable improvement in materials properties at the different stages of polymerizations compared to virgin and/or standard materials. At this point, like to leave the decision on acceptance of the manuscript at the discretion of the journal editors.